# Estimation of the Periodontal Inflamed Surface Area by Simple Oral Examination

**DOI:** 10.3390/jcm10040723

**Published:** 2021-02-12

**Authors:** Yoshiaki Nomura, Toshiya Morozumi, Yukihiro Numabe, Yorimasa Ogata, Yohei Nakayama, Tsutomu Sugaya, Toshiaki Nakamura, Soh Sato, Shogo Takashiba, Satoshi Sekino, Nobuo Yoshinari, Nobuhiro Hanada, Naoyuki Sugano, Mitsuo Fukuda, Masato Minabe, Makoto Umeda, Koichi Tabeta, Keiso Takahashi, Kazuyuki Noguchi, Hiroaki Kobayashi, Hideki Takai, Fusanori Nishimura, Fumihiko Suzuki, Erika Kakuta, Atsutoshi Yoshimura, Atsushi Saito, Taneaki Nakagawa

**Affiliations:** 1Department of Translational Research, Tsurumi University School of Dental Medicine, Yokohama 230-8501, Japan; nomura-y@tsurumi-u.ac.jp (Y.N.); hanada-n@tsurumi-u.ac.jp (N.H.); 2Division of Periodontology, Department of Oral Interdisciplinary Medicine, Graduate School of Dentistry, Kanagawa Dental University, Yokosuka 238-8580, Japan; minabe@kdu.ac.jp; 3Department of Periodontology, School of Life Dentistry at Tokyo, The Nippon Dental University, Tokyo 102-8159, Japan; numabe-y@tky.ndu.ac.jp (Y.N.); sekino-s@tky.ndu.ac.jp (S.S.); 4Department of Periodontology, Nihon University School of Dentistry at Matsudo, Matsudo 271-8587, Japan; nakayama.youhei@nihon-u.ac.jp (Y.N.); ogata.yorimasa@nihon-u.ac.jp (Y.O.); takai.hideki@nihon-u.ac.jp (H.T.); 5Division of Periodontology and Endodontology, Department of Oral Health Science, Hokkaido University Graduate School of Dental Medicine, Sapporo 060-8586, Japan; sugaya@den.hokudai.ac.jp; 6Department of Periodontology, Kagoshima University Graduate School of Medical and Dental Sciences, Kagoshima 890-8544, Japan; toshi-n@dent.kagoshima-u.ac.jp (T.N.); kazuperi@dent.kagoshima-u.ac.jp (K.N.); 7Department of Periodontology, School of Life Dentistry at Niigata, The Nippon Dental University, Niigata 951-8580, Japan; s-sato@ngt.ndu.ac.jp; 8Department of Pathophysiology-Periodontal Science, Okayama University Graduate School of Medicine, Dentistry and Pharmaceutical Sciences, Okayama 700-8525, Japan; stakashi@okayama-u.ac.jp; 9Department of Periodontology, School of Dentistry, Matsumoto Dental University, Shiojiri 399-0781, Japan; nobuo.yoshinari@mdu.ac.jp; 10Department of Periodontology, Nihon University School of Dentistry, Tokyo 101-8310, Japan; sugano.naoyuki@nihon-u.ac.jp; 11Department of Periodontology, School of Dentistry, Aichi Gakuin University, Nagoya 464-8650, Japan; fukuda-m@dpc.agu.ac.jp; 12Department of Periodontology, Osaka Dental University, Hirakata 573-1121, Japan; umeda-m@cc.osaka-dent.ac.jp; 13Division of Periodontology, Department of Oral Biological Science, Niigata University Graduate School of Medical and Dental Sciences, Niigata 951-8514, Japan; koichi@dent.niigata-u.ac.jp; 14Division of Periodontics, Department of Conservative Dentistry, Ohu University School of Dentistry, Koriyama 963-8611, Japan; ke-takahashi@den.ohu-u.ac.jp; 15Department of Periodontology, Graduate School of Medical and Dental Sciences, Tokyo Medical and Dental University, Tokyo 113-8510, Japan; h-kobayashi.peri@tmd.ac.jp; 16Section of Periodontology, Division of Oral Rehabilitation, Faculty of Dental Science, Kyushu University, Fukuoka 812-8582, Japan; fusanori@dent.kyushu-u.ac.jp; 17Division of Dental Anesthesiology, Department of Oral Surgery, Ohu University School of Dentistry, Koriyama 963-8611, Japan; f-suzuki@den.ohu-u.ac.jp; 18Department of Oral Microbiology, Tsurumi University School of Dental Medicine, Yokohama 230-8501, Japan; kakuta-erika@tsurumi-u.ac.jp; 19Department of Periodontology and Endodontology, Nagasaki University Graduate School of Biomedical Sciences, Nagasaki 852-8588, Japan; ayoshi@nagasaki-u.ac.jp; 20Department of Periodontology, Tokyo Dental College, Tokyo 101-0061, Japan; atsaito@tdc.ac.jp; 21Department of Dentistry and Oral Surgery, School of Medicine, Keio University, Tokyo 160-8582, Japan; tane@z6.keio.jp

**Keywords:** periodontal diseases, periodontal pocket, health status indicators, periodontal inflamed surface area (PISA), item response theory

## Abstract

The periodontal inflamed surface area (PISA) is a useful index for clinical and epidemiological assessments, since it can represent the inflammation status of patients in one contentious variable. However, calculation of the PISA is difficult, requiring six point probing depth measurements with or without bleeding on probing on 28 teeth, followed by data input in a calculation program. More simple methods are essential for screening periodontal disease or in epidemiological studies. In this study, we tried to establish a convenient partial examination method to estimate PISA. Cross-sectional data of 254 subjects who completed active periodontal therapy were analyzed. Teeth that represent the PISA value were selected by an item response theory approach. The maxillary second molar, first premolar, and lateral incisor and the mandibular second molar and lateral incisor were selected. The sum of the PISAs of these teeth was significantly correlated with the patient’s PISA (R^2^ = 0.938). More simply, the sum of the maximum values of probing pocket depth with bleeding for these teeth were also significantly correlated with the patient’s PISA (R^2^ = 0.6457). The simple model presented in this study may be useful to estimate PISA.

## 1. Introduction

Several periodontal indexes have been proposed and used for clinical or epidemiological assessments. The indexes used in epidemiological studies include the commonality periodontal index (CPI) [1,2,3], clinical attachment level (CAL) [4,5], and bleeding on probing (BOP) [6], while other studies used combinations of indexes, such as CPI, CAL, BOP, and probing pocket depth (PD) [7], CAL, PD, gingival index (GI), and plaque index (PlI) [8], CAL, PD, and GI [9], and CAL and BOP [10]. For clinical assessments, PD, CAL, and BOP have been commonly used for evaluation of diagnoses or therapeutic effects. Although summary statistics of these parameters, like the mean or maximum value, have been conventionally used, these summary statistics involve loss of information. The periodontal inflamed surface area (PISA) was developed to address these issues [11]. PISA is a very convenient index that reflects the surface area of bleeding pocket epithelium in square millimeters and is calculated using conventional clinical parameters of periodontal health, namely BOP combined with either PD, or CAL and gingival recession [11]. Thus, PISA can represent the inflammation status of subjects in one contentious variable, for which calculation of mean or maximum value is not necessary. Recent study have reported that PISA is effectively associated with systemic markers of low-grade inflammation, such as C-reactive protein [12]. Periodontal disease is a risk factor for non-communicable diseases through the localized inflammation and periodontal pathogens [13,14,15,16,17,18,19,20]. Therefore, PISA may be an optimal index to investigate the correlation between periodontal disease and non-communicable diseases.

However, calculation of PISA is an extremely difficult task, requiring six probing depth measurements of all the teeth and BOP and PD data for 168 sites. A simpler method is indispensable for screening or epidemiological studies. Therefore, in this study, we tried to establish a convenient partial examination method to estimate PISA. Our simple, convenient, and evidence-based partial oral examination method to estimate PISA may be a useful tool like other periodontal indexes for screening or epidemiological studies.

## 2. Materials and Methods

### 2.1. Study Design

#### 2.1.1. Setting

This study was a part of a clinical research project conducted by the Japanese Society of Periodontology in cooperation with 17 facilities (one clinic and 16 university hospitals) in Japan for the diagnosis of periodontitis. In our previous reports, we had analyzed 124 participants who successfully completed the study protocol [21,22,23]. For this study, we selected 254 patients with chronic periodontitis who had completed their active treatment under the regulations of the Japanese health insurance system. After registering for a screening examination before their follow-up, all 254 patients were analyzed in the study. The inclusion criteria were age greater than 30 years, number of remaining teeth more than 20, and systemically healthy status.

#### 2.1.2. Diagnosis

Each patient was diagnosed according to the guidelines used at the time (Guidelines of the American Academy of Periodontology) [24]. Oral examinations were carried out by one examiner at each institute (T.M., Y.N., T.S., T.N., S.S., S.S., N.Y., N.S., M.F., M.M., K.N., H.K., H.T., F.S., A.Y., and T.N.). Each examiner was a periodontist licensed by the Japanese Society of Periodontology. Intra- and interexaminer calibration sessions were conducted using periodontal disease models (P15FE-500HPRO-S2A1-GSF, NISSIN, Kyoto, Japan) at the beginning and middle of the study period. In brief, full-mouth PD and recessions were measured twice, and repeatability for CAL was assessed. The examiner was judged to have made reproducible measurements after reaching a percentage of agreement within ± 1 mm between repeated measurements of at least 95% of measurements.

### 2.2. Research Data

In this study, we analyzed PD and BOP data for calculation of PISA. A freely downloadable spreadsheet is available to calculate the PISA [11], and CAL and PlI were used as associated factors for PISA. Details of the data are described in our previous reports [21,22,23].

### 2.3. Statistical Analysis

The number of bleeding sites and PDs were measured at six sites for all of the remaining teeth with a periodontal probe (CP-12 Color-Coded Probe; Hu-Friedy, Chicago, IL, USA). To find out the optimal value for IRT analysis, 6 kinds of cut-off points were set for number of bleeding site per teeth: (0, 1–6), (0 and 1, 2–6), (0–2, 3–6), (0–3, 4–6), (0–4, 5 and 6), and (0–5, 6). For each cut-off point, models were constructed. For the PD, the maximum value (mm) for one tooth in the six sites were used.

By using the dichotomized data, a three-parameter logistic model based on the item response theory (IRT) was applied. For the IRT analysis, R software with the ltm and irtoys package was used.

### 2.4. Ethical Approval

The study was conducted in compliance with the principles outlined in the Helsinki Declaration. Written informed consent was obtained from each participant, and the protocol was approved by the Institutional Review Board of each participating institution. The ethics committee members’ names and reference numbers are as follows: The regional ethical committee of the Faculty of Dentistry, Niigata University (20-R17-08-06); Keio University School of Medicine, Ethics Committee (20080096); Ethical Committee of Kagoshima University Medical and Dental Hospital (20-58); The Ethics Committee, Nagasaki University Graduate School of Biomedical Sciences (0846-2); Ethical Committee of Kyushu University Faculty of Dental Science (20-11); The Ethics Committee of Osaka Dental University (80712); Ethics Committee of Aichi Gakuin University, School of Dentistry (158); The Ethics Committee of Matsumoto Dental University (0090); The Institutional Review Board of Nippon Dental University (2-1-22); Ethical Committee of Nihon University School of Dentistry (EP08D016); Dental Research Ethics Committee of Tokyo Medical and Dental University (660); Ethics Committee in Nihon University School of Dentistry at Matsudo (EC 08-014); Ethics Committee of Tokyo Dental College (208); The Ethical Review Committee of The Nippon Dental University School of Life Dentistry at Niigata (151); Ohu University Research Ethics Committee (52); Institutional Review Board for Clinical Research of Hokkaido University Hospital (008-0113).

## 3. Results

### 3.1. Participant Characteristics

The study population consisted of 114 men and 140 women. Their mean age was 55.6 +/− 10.3 years, and their mean number of remaining teeth was 25.4 +/− 2.61.

### 3.2. Prediction of PISA by IRT Analysis Based on the Number of BOP Sites

Since PISA is calculated on the basis of BOP and PD measurements, to reduce the number of BOP measurements, IRT analysis was performed. Cut-off points were set as at least 1, 2, 3, and 4 sites for BOP. The ability, which indicates the weighted sum of the total number of bleeding sites, was calculated for each participant. Scatter plots were obtained for the PISA against the ability calculated by IRT analysis. After log_10_ transformation of PISA, linear relationships were observed. The results are shown in Figure 1, and the models are shown in Appendix A. Item response curves and item information curves are shown in Appendix A.

The results indicated that at least one site with bleeding for each tooth may be enough for evaluation of bleeding on probing.

### 3.3. Prediction of PISA by the Maximum Value of the PD at Each Tooth by IRT Analysis

In clinical settings, even the calculation of mean values of PD may be laborious. Thus, assessments based on the maximum value are simpler and more suitable for rapid evaluations. Therefore, the maximum PD value for each tooth was used as the variable. To investigate which site represents the PISA, a three-parameter logistic model based on IRT was used. To transform the maximum values for dichotomous variable, cut-off values were set as >3 mm, >4 mm, 5 mm, and 6 mm. The results are shown in Figure 2, the models are indicated in Appendix A, and the item response curves and item information curves are shown in Appendix A.

### 3.4. Prediction of PISA Based on the Selected Sites

The item information curves in Appendix A provided valuable information for tooth selection. The area under the information curves for each tooth were calculated, and the results were presented by the mean values for left and right teeth (Table 1). Teeth with relatively higher information were the maxillary second molar, maxillary first premolar, maxillary lateral incisor, mandibular second molar, and mandibular lateral incisors. To confirm that the selected teeth were optimal, scatterplots of the overall PISA against the PISAs of these teeth were illustrated. The results are presented in Appendix A. R^2^ values of these teeth were relatively higher.

### 3.5. Prediction of PISA Based on the Selected Teeth

#### 3.5.1. Correlation of PISA of Selected Teeth with PISA

On the basis of the data presented in Table 1 and Appendix A, five teeth were selected: the maxillary second molar, maxillary first molar, maxillary lateral incisor, mandibular second molar, and mandibular lateral incisor. The scatter plot for the PISA against the sum of PISAs of these 10 teeth are provided in Figure 3. Since linear regression is affected by outliers, PISA less than 1000 is illustrated again in Figure 3B. The R^2^ was 0.938 for all the data and 0.817 for PISA less than 1000.

#### 3.5.2. Prediction of PISA by the Maximum Value of PD

To create a simple model for clinical convenience, the sum of the maximum PD values with BOP were used to generate a summary score to predict PISA. The scatter plot with the net values showed an exponential curve (Figure 4A). When the values were log_10_ transformed, a linear relationship was observed. The PISA against a value of 0 on the *X*-axis was less than 2. This indicated that when the 10 teeth did not show any bleeding, PISA is less than 100.

The effect of missing teeth is uncertain. In addition, a selected site of 0 may not guarantee a PISA value of 0. Figure 4 was categorized by the groups; Group 0: no missing teeth and no bleeding in 10 teeth, Group 1: one missing teeth and no bleeding in nine teeth, Group 2: two missing teeth and no bleeding in eight teeth, Group 3: three missing teeth and no bleeding in seven teeth, Group 10: one missing teeth in 10 teeth, Group 11: one missing teeth in 10 teeth, Group 12: two missing teeth in 10 teeth, Group 13: three missing teeth in ten teeth, Group 14: four or five missing teeth in ten teeth (Appendix A).

If there was no bleeding in ten teeth, PISA was estimated a less than 100. Thus, there may be little effect of missing teeth on PISA (Figure 4B).

The final model is presented in the following formula:Log_10_ (PISA) = 1.592 × log_10_Σ(maximum value of probing depth withbleeding of selected teeth) + 0.4111(1)

Selected teeth are maxillary second molar, maxillary first premolar, maxillary lateral incisor, mandibular second molar, and mandibular lateral incisor.

When separately predicted PISA by bleeding of approximal surface and flat surface, effects of bleeding in approximal surface was larger than flat surface. The results are shown in Appendix A.

Additionally, for clinical convenience, quick reference for the predictive values of PISA calculated by the formula described above is presented in Appendix A.

## 4. Discussion

In this study, we tried to establish a partial examination method using the selected teeth to estimate PISA. The final simple model that uses the sum of the maximum PD values of the 10 selected teeth with BOP may be useful to estimate PISA.

Many studies have used PISA for evaluation of periodontal conditions. Since PISA reflects the inflammation status of periodontal tissue, these studies focused on the correlation of PISA and systemic diseases [25,26,27,28,29,30,31,32,33,34,35,36,37,38,39,40,41,42,43], or the correlation of PISA with novel disease markers [41,42,43,44,45,46,47,48]. In the present study, IRT analysis was used to set a cut-off point for the number of bleeding sites and the maximum values of PD in a single tooth. IRT is based on the relationship between the performance of the subjects on a test item and the overall measured ability. It can calculate the weight of each item and the ability of individuals. These values correspond to the weight of clinical symptoms of each teeth and PISA. Therefore, selection of the teeth as items and calculation of the ability as PISA with IRT is a reasonable approach.

As shown in Figure 1, the cut-off point of one site of bleeding in one tooth may be suitable. When the cut-off point was set to more than two sites, the number of participants with an estimated PISA of 0 increased. The number of participants who missed with less than 100 PISA was also increased. If the PD is transformed to a dichotomous variable by an optimal cut-off point, several statistical methods for the screening such as ROC analysis and evaluations of sensitivity, specificity, positive predictive value, and negative predictive value become available. However, the optimal cut-off points were not presented by previous studies. In addition, the scatterplots shown in Figure 2 were not as clear as the scatter plot shown in Figure 1A. Therefore, in the model shown in Figure 4, a simple sum of the maximum value of PD in the selected teeth with BOP was applied as an independent value. However, this simple model was sufficient to estimate PISA. Missing teeth had little effect on PISA in the simple model shown in Figure 4. Since PISA is a sum of the PISAs of each tooth, a value of 0 as a result of missing teeth would not include the PISA. However, the simple model could not detect participants with PISA less than 100.

The PISA value for the diagnosis of periodontal disease has been defined previously [48]. However, calculation of PISA is more laborious than examination of periodontal tissue for the diagnosis. The value of PISA was correlated with the clinical parameters of periodontal tissue and periodontal pathogens [49] and dental plaque metabolic byproducts [50]. Only a few studies have investigated the characteristics of PISA. Thus, additional studies are necessary to understand the characteristics of PISA.

Representative six teeth by Ramfjord was used in epidemiological studies. Although it reflect the entire periodontal disease, those sites were selected from studies by numerous investigators using (a) P.M.A. index, (b) formation of pocket depth and bone loss, and (c) teeth extraction record [51]. While PISA focus on degree of periodontal inflammation, as an advantage of the study, we calculated using our raw data for selection.

There is a limitation of this study. Study population of this study was consisted of the patients who finished periodontal treatment. There may be bias in periodontal conditions. Further study is necessary to confirm the availability the model presented in this study by epidemiological study which included the periodontal healthy subjects and subjects with severe periodontal conditions. In addition, further study is necessary to confirm application of the model presented in this study for the patients with systemic diseases.

In this study, by using patients’ data, we tried to establish a convenient partial examination method to estimate PISA. Simple, convenient, and evidence-based partial examination methods to estimate PISA may be useful tools for the screening or epidemiological studies, such as other periodontal indexes.

## Figures and Tables

**Figure 1 jcm-10-00723-f001:**
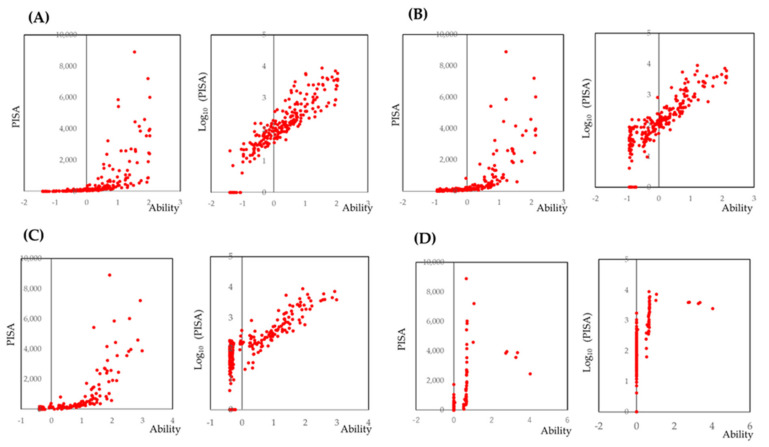
Scatter plots of PISA against the number of bleeding sites within each tooth. (**A**–**D**) Scheme 1. 2, 3, and 4 bleeding sites within each tooth. Ability, which indicates the weighted sum of the number of sites with bleeding on probing, was calculated by the three-parameter logistic model based on item response theory analysis. When PISA was log_10_ transformed, linear relationships were observed.

**Figure 2 jcm-10-00723-f002:**
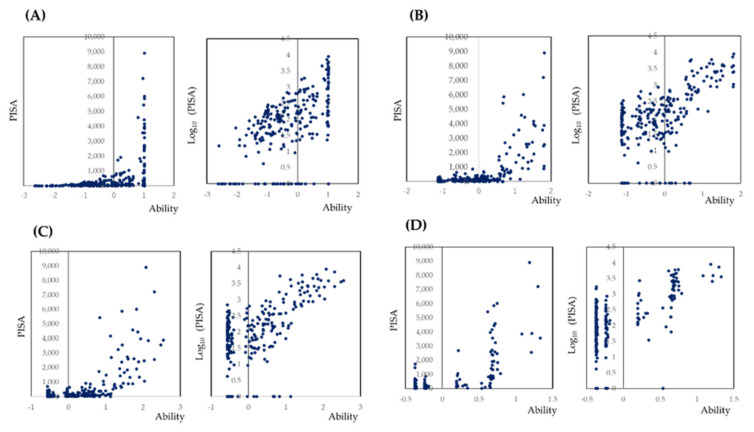
Scatter plots of PISA against the maximum value of PD within each teeth. (**A**–**D**) shows the findings with cut-off points of 3, 4, 5, and 6 mm of PD within one tooth. When PISA was log_10_ transformed, linear relationships were observed.

**Figure 3 jcm-10-00723-f003:**
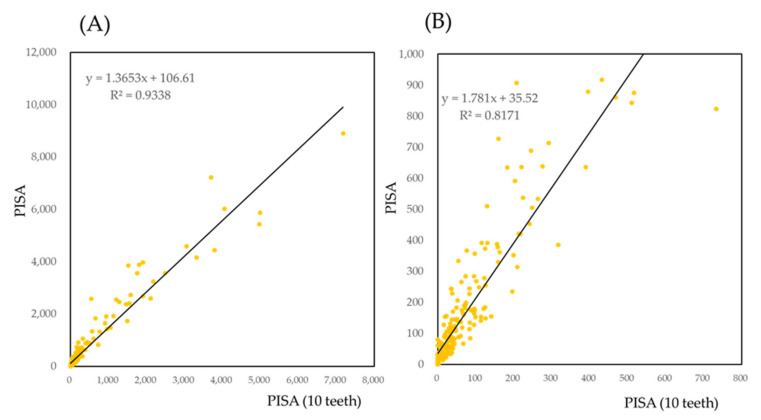
Scatter plot of PISA against the maximum value of PD in each tooth. (**A**,**B**) are the same figure. (**B**) is magnified PISA for less than 1000. Regression line (**B**) was calculated by PISA less than 1000.

**Figure 4 jcm-10-00723-f004:**
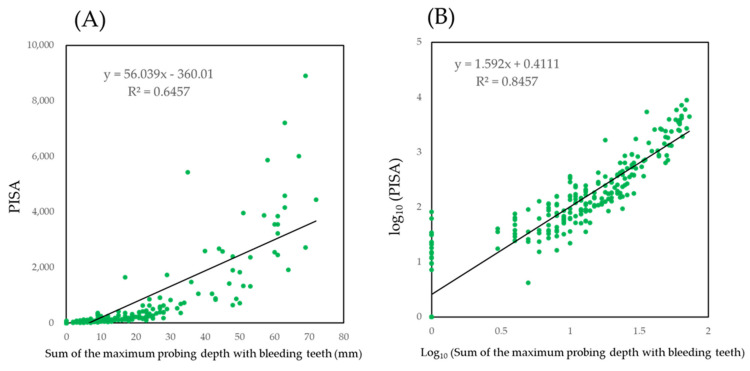
Scatter plot of PISA against sum of the maximum values of PD for each of the bleeding teeth. (**A**) shows the scatter plot by net values. (**B**) shows the values were log_10_ transformed.

**Table 1 jcm-10-00723-t001:** Item information for tooth type for the number of bleeding sites and the maximum value of probing depth.

		Bleeding on Probing			Maximum Value of Probing Depth		
		1 Site	2 Sites	3 Sites	3 Mm	4 mm	5 mm
Maxillary	2nd Molar	27.0	35.9	128.7	22.4	29.9	32.1
	1st Molar	25.4	23.3	35.9	24.0	35.9	43.8
	2nd Premolar	30.7	32.3	47.0	35.1	41.0	47.8
	1st Premolar	33.7	39.9	53.8	30.5	40.1	43.7
	Canine	28.5	35.1	48.0	32.2	43.5	429.5
	Lateral incisor	27.8	48.4	71.7	30.8	44.8	49.8
	Central incisor	30.7	32.1	42.4	28.7	38.5	41.4
Mandibular	2nd Molar	28.0	27.3	36.6	24.6	34.5	30.5
	1st Molar	27.8	27.6	39.1	15.0	37.7	47.6
	2nd Premolar	26.7	27.2	32.8	32.3	34.6	31.7
	1st Premolar	32.6	27.9	33.6	31.1	43.9	39.0
	Canine	24.8	30.3	35.1	35.3	47.7	51.6
	Lateral incisor	32.0	32.6	32.6	43.1	48.3	41.3
	Central incisor	36.8	32.7	34.8	42.8	46.2	35.5

## Data Availability

The data presented in this study are available on request from the corresponding author. The data are not publicly available due to privacy.

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
