# Peer review of "Estimation of the Periodontal Inflamed Surface Area by Simple Oral Examination"

_jcm, 2021, doi:10.3390/jcm10040723_

Round 1

Reviewer 1 Report

Title: Estimation of the periodontal inflamed surface area by simple oral examination.

Manuscript ID: jcm-1084384

Some clarifications and modifications should be considered to improve the manuscript.

Key Words: None of the proposed key words are MeSH terms. Include correct key words so that if the article is accepted, it can be correctly located in the bibliographic repertories (PubMed).

Introduction.

-The authors state in the introduction: "These indexes are based on the partial oral examination of the periodontal tissue", referring to CAL, PPD, BoP, PI. This is not correct, the vast majority of periodontal studies assess the periodontal clinical status of all teeth present in the mouth, and not partially.

-The authors state in the introduction: "PISA is a very convenient index that reflects the perinatal inflamed status". what does this mean? Did you mean periodontal?.

M&M

Study Desing.

- It is unclear when they performed PD and BoP data determinations to calculate PISA. these variables appear to have been determined at a periodontal maintenance appointment. If this is so, there is an evident bias on the periodontal status of these patients. If not, and they were assessed before the initial anti-inflammatory treatment, exclusion criteria such as the intake of anti-inflammatory drugs and / or AB, had to have been considered.

Diagnosis.

- The authors state:  Each patient was diagnosed according to the guidelines used at the time (Guidelines of the American Academy of Periodontology) (29). Citation 29 does not correspond to the APA Guide Lines, please correct it.

-Since the examiners were calibrated, why don't they show agreement data or Kappa values?.

-What periodontal probe was used for diagnosis?

Statistical analysis.

-This reviewer does not understand how the authors describe as dichotomous values, the 6 data that express BOP and the single data, for PD.

Ethical approval.

-The reference number of the approval of the Ethics Committee must appear in the manuscript, and not as an appendix.

Discussion

-The authors state "the cut-off point of 1 site of bleeding in one tooth may be suitable. In the recent classification of periodontal diseases, the AAP and FEP (published in 2018), they advise measuring BOP at six sites per tooth. This criteria is not insignificant, since exclusive interproximal bleeding has a different clinical significance than bleeding from free surfaces. In another sense, there are classically described groups of teeth representative of the entire mouth, as is the case of the 6 teeth of Ramfjord  (Indices for prevalence and incidence of periodontal disease. J Periodontol 1959; 30: 51-9) that could be used in epidemiological studies. What are the advantages of this study? These aspects should be sufficiently discussed.

References

-Is the self-citation of all the authors' previous works necessary? Unnecessary self-citation can be inappropriate behavior.

Author Response

Response to Reviewer 1’s Comments

Thank you for your valuable comments to improve our manuscript. We reply your comments point by point. The changes according to your comments highlighted yellow (Sky-blue: reviewer 2).

Keywords

Question 1): None of the proposed key words are MeSH terms. Include correct key words so that if the article is accepted, it can be correctly located in the bibliographic repertories (PubMed).

Response: Thank you for your indication. We corrected “periodontal disease index” to “Periodontal Index [MeSH]” (page 2, line 77).

Introduction

Question 2): The author state in the introduction: "These indexes are based on the partial oral examination of the periodontal tissue", referring to CAL, PPD, BoP, PI. This is not correct, the vast majority of periodontal studies assess the periodontal clinical status of all teeth present in the mouth, and not partially.

Response: Thank you for your indication. We agree with your opinion. The sentence is removed (page 2, line 86).

Question 3): The author state in the introduction: "PISA is a very convenient index that reflects the perinatal inflamed status". What does this mean? Did you mean periodontal?.

Response: Thank you for your indication. It was misspelling. We corrected the sentence (page 2, line 91-93).

Materials and Methods

Study Design

Question 4): It is unclear when they performed PD and BOP data determinations to calculate PISA. These variables appear to have been determined at a periodontal maintenance appointment. If this is so, there is an evident bias on the periodontal status of these patients. If not, and they were assessed before the initial anti-inflammatory treatment, exclusion criteria such as the intake of anti-inflammatory drugs and / or AB, had to have been considered.

Response: Thank you for your indication. As described in Materials and Method, study population consisted of the patients who finished the periodontal treatment. They were not random sample from general population. Therefore, there may be bias in periodontal conditions. We described it in Discussion as limitation of this study (page 8, line 279-282).

Diagnosis

Question 5): The author state: Each patient was diagnosed according to the guidelines used at the time (Guidelines of the American Academy of Periodontology) (29). Citation 29 does not correspond to the APA Guide Lines, please correct it.

Response: Thank you for your indication. We corrected the misdescription from citation 29 to 24 (page 3, line 120).

Question 6):  Since the examiners were calibrated, why don't they show agreement data or Kappa values?

Response: Thank you for your indication. We described the calibration in the manuscript (page 3, line 123-127).

Question 7): What periodontal probe was used for diagnosis?

Response: Thank you for your indication. We described the probe we used in the study (page 3, line 134).

Statistical analysis.

Question 8): This reviewer does not understand how the authors describe as dichotomous values, the 6 data that express BOP and the single data, for PD.

Response: Thank you for your indication. We modified the description to be more specific. (page 3, line 134-137).

Ethical approval

Question 9): The reference number of the approval of the Ethics Committee must appear in the manuscript, and not as an appendix.

Response: Thank you for your advice. We contained a written description of the reference number of the approval of the Ethics Committee in the manuscript (page 4, line 148-160).

Discussion

Question 10): The authors state "the cut-off point of 1 site of bleeding in one tooth may be suitable. In the recent classification of periodontal diseases, the AAP and EFP (published in 2018), they advise measuring BOP at six sites per tooth. This criteria is not insignificant, since exclusive interproximal bleeding has a different clinical significance than bleeding from free surfaces. In another sense, there are classically described groups of teeth representative of the entire mouth, as is the case of the 6 teeth of Ramfjord (Indices for prevalence and incidence of periodontal disease. J Periodontol 1959; 30: 51-9) that could be used in epidemiological studies. What are the advantages of this study? These aspects should be sufficiently discussed.

Response: Thank you for your suggestion. According to your advice, we analyzed the correlation of free surface and interproximal bleeding. Results of the item response curve is shown in following Figure. Interproximal bleeding and buccal free surface show similar behavior.

When calculated the prevalence of bleeding, most of the teeth were not without bleeding. Three point 4 percent tooth had only flat surface bleeding and 22.4% teeth had only approximal surface bleeding.

Flat surface

Total

-

+

Approximal surface

-

3933(61.0%)

222(3.4%)

4155(64.5%)

+

1446(22.4%)

842(13.1%)

2288(35.5%)

Total

5379(83.5%)

1064(16.5%)

6443(100%)

Therefore, predictions of PISA value were separately carried out by bleeding in approximal surface and flat surface. Results were shown in Figure S5 and following sentence was inserted in Results (page 7, line 238-239).

“When separately predicted PISA by bleeding of approximal surface and flat surface, effects of bleeding in approximal surface was larger than flat surface. The results were shown in Figure S5.”

Figure S5. Scatter plot of PISA against the PISA calculated by bleeding with approximal surface (A) and flat surface (B)

Similarity, prediction of PISA was carried out by the at least bleeding of approximal surface or flat surface in one site with in one teeth. When compared R2 of the regressions, R2 of regression by approximal surface was higher than that of flat surface. It indicates that the effect for PISA value by approximal surface was higher than that of flat surface.

In addition, representative 6 teeth by Ramfjord used in epidemiological studies represent the entire periodontal disease. However, those sites were selected from studies by numerous investigators using a ) P.M.A. index, b) formation of pocket depth and bone loss, and c) teeth extraction record. While, PISA focus on degree of periodontal inflammation, as an advantage of the study, we calculated using our raw data for selection (page 8, line 274-278)

References

Question 11): Is the self-citation of all the authors' previous works necessary? Unnecessary self-citation can be inappropriate behavior.

Response: Thank you for your advice. References concerning IRT analysis were minimized. We removed Reference 25-30 in previous manuscript.

Reviewer 2 Report

The introduction could be improved explaining better what is the PISA and how is normally calculated.

Inclusion criteria of this study are too much stricter. Periodontitis is often diagnosed in not healthy patients suffering from diabetes, immune-deficiencies, stress, hormonal imbalances or patients undergoing therapies associated with dry mouth or gingival hyperplasia. So the evaluation of only healthy patients could represent a bias for the study. 

This is a multicentre study with its pros and cons. Surely it can results in higher rates of patient enrollment than single-centre, but at the same time, the presence of one examiner per each institute (so 17 different facilities) could represent a bias with heterogeneity of the measurement.

The final "simple" model is a logarithmic formula that could be difficult to apply in the daily clinical practice.  

Author Response

Response to Reviewer 2’s Comments

Thank you for your valuable comments to improve our manuscript. We reply your comments point by point. The changes according to your comments highlighted sky-blue (yellow: reviewer 1).

Question 1):  The introduction could be improved explaining better what is the PISA and how is normally calculated.

Response: Thank you for your suggestion. We described them in the Introduction (page 2, line 90-96). Values of PISA is calculated by BOP combined with either PD, or CAL and gingival recession. PISA is calculated by six degree calculation formula. For instance, PISA of maxillary 1st and 2nd molar is calculated as follow:

Maxillary 2nd molar

{25.4265✕(Mean PD) +4.6241✕(Mean PD)2 + 3.0787✕ (Mean PD)3 + 0.06519✕ (Mean PD)4-0.10923✕ (Mean PD)5 + 0.0040876✕ (Mean PD)6}✕(Number of bleeding site)

Maxillary 1st molar

{16.8835✕(Mean PD) - 0.5688✕(Mean PD)2 + 1.5433✕ (Mean PD)3 - 0.06519✕ (Mean PD)4-0.0145✕ (Mean PD)5 + 0.0009019✕ (Mean PD)6}✕(Number of bleeding site)

We can show 14 kinds of these formula by tooth type. Calculation is complex. Therefore, Japanese society of periodontology provided free EXCEL spread sheet in the Home page. Original article of PISA provided freely available calculation spread sheet (website: http://www.parsprototo.info.)

These formulas are not our original. We hesitated to describe the formula even in supplemental materials. Originality is in Nesses W et al. We think precise description of the formula in our manuscript is out of rule.

Question 2):  Inclusion criteria of this study are too much stricter. Periodontitis is often diagnosed in not healthy patients suffering from diabetes, immune-deficiencies, stress, hormonal imbalances or patients undergoing therapies associated with dry mouth or gingival hyperplasia. So the evaluation of only healthy patients could represent a bias for the study.

Response: Thank you for your indication. As you pointed out, there are many periodontitis patients with systemic illness. While, we dared to recruit periodontitis patients without systemic disease because the effect of systemic inflammation can be a bias for PISA.

Question 3): This is a multicentre study with its pros and cons. Surely it can results in higher rates of patient enrollment than single-centre, but at the same time, the presence of one examiner per each institute (so 17 different facilities) could represent a bias with heterogeneity of the measurement.

Response: Thank you for your indication. We performed intra- and inter-examiner calibration sessions at the beginning and middle of the study period to avoid bias. We described it in the manuscript (page 3, line 123-127).

Question 4): The final "simple" model is a logarithmic formula that could be difficult to apply in the daily clinical practice. 

Response: Thank you for your indication. For the clinical convenience, quick reference was inserted as Table S3, and following sentence was inserted in Table S3. Additionally, for clinical convenience, quick reference for the predictive values of PISA calculated by the formula described above is presented in Table S3 (page 7, line 240-241).

Reviewer 3 Report

Comments to the Author I read very interesting and novel research article with the title "Estimation of the periodontal inflamed surface area by simple oral examination". The aim of the study is novel, impressive and beneficial for the dental society. The manuscript is novel, scientifically sound and well written, thus can be accepted after some minor corrections. Still more discussion on practical applications and future perspectives of the study is essential. I understood that the PISA methods may be useful tools for the screening or epidemiological studies but still isn't clear how simple it can be for each dentist to estimate of the periodontal inflamed surface area by simple oral examination. It seems that this method is very laborious and difficult. Please try to comment it in clear way in the discussion section.

Author Response

Response to Reviewer 3’s Comments

Thank you for your valuable comments to improve our manuscript. We reply your comments point by point.

Question 1):  Still more discussion on practical applications and future perspectives of the study is essential.

Response: Thank you for your indication. In clinical practice, oral examination should be precisely carried out for all the teeth. However, every time patients visit the dental clinic, it is impossible to carry out precise oral examination every time. In addition, periodontal inflammation considered to be one of the important etiology of several diseases. In facts, most of the study concerning PISA was to instigate the correlation between PISA and incidence of systemic diseases. Measurement of PISA is a laborious work. Measurement and recording probing pocket depth and bleeding on probing of six site of all the teeth tales time. In addition, data entry and calculation is necessary. In other words, results cannot obtain until all these process is completed. It makes the clinical application of PISA is difficult. To obtain sophisticated evidence of the correlation between periodontal disease and systemic disease, epidemiological study with large sample size is in dispensable. For this purpose, PISA is a useful tool, however, application of PISA for the large sample is difficult. The model presented in this study makes the measurement and calculation of PISA simple. Quick reference presented in Table S3 makes the calculation simple.  

Question 2): I understood that the PISA methods may be useful tools for the screening or epidemiological studies but still isn't clear how simple it can be for each dentist to estimate of the periodontal inflamed surface area by simple oral examination. It seems that this method is very laborious and difficult. Please try to comment it in clear way in the discussion section.

Response: The oral examination and calculation should be more concise. However, succinct method makes loss of information. The availability of the results presented in this study is to make the effort minimum. Other conventional periodontal indexes need to examine several tooth. The model presented in this study also needs examine several teeth. The difference of the model presented in this study with other conventional periodontal index is that the gold standard asset as PISA which is recently recommended to use an index that reflects the inflammation statues of periodontal tissue.

Round 2

Reviewer 1 Report

Most of the questions from the first review have been properly answered and / or corrected. Just two more questions:

-The authors could justify why there are still key words that are not MeSH terms? Does it have any purpose?

- One issue that has been ignored, is the self-citation of all the authors. As I pointed out from unnecessary self-citation can be inappropriate behavior. Can the authors explain how citations like 25, about Caries in Primary Dentition in Myanmar Children, are related to this manuscript? 

Author Response

Response to Reviewer 1’s Comments

Thank you for your valuable comments to improve our manuscript. We reply your comments point by point. The changes according to your comments highlighted yellow (Sky-blue: reviewer 2).

Question 1): The authors could justify why there are still key words that are not MeSH terms? Does it have any purpose?

Response: Thank you for your indication. We removed “Periodontal Index”, and added adequate 3 keywords (“Periodontal Diseases”, “Periodontal Pocket”, “Health Status Indicators”).

Question 2): One issue that has been ignored, is the self-citation of all the authors. As I pointed out from unnecessary self-citation can be inappropriate behavior. Can the authors explain how citations like 25, about Caries in Primary Dentition in Myanmar Children, are related to this manuscript?

Response: Thank you for your appropriate indication. We removed no. 25 from references.

Reviewer 2 Report

Response: Thank you for your suggestion. We described them in the Introduction (page 2, line 90-96).

The reviewer appreciates this clarification.

Response: Thank you for your indication. As you pointed out, there are many periodontitis patients with systemic illness. While, we dared to recruit periodontitis patients without systemic disease because the effect of systemic inflammation can be a bias for PISA.

So the authors believe that their modified form of PISA could be applied and used to patients with systemic illness? or in patients with systemic inflammation is not possible to apply?

Response: Thank you for your indication. We performed intra- and inter-examiner calibration sessions at the beginning and middle of the study period to avoid bias. We described it in the manuscript (page 3, line 123-127).

The reviewer appreciates this clarification. 

Author Response

Response to Reviewer 2’s Comments

Thank you for your valuable comments to improve our manuscript. We reply your comments point by point. The changes according to your comments highlighted sky-blue (yellow: reviewer 1).

Question 1):  So the authors believe that their modified form of PISA could be applied and used to patients with systemic illness? or in patients with systemic inflammation is not possible to apply?

Response: Thank you for your indication. We inserted following sentence in limitations (page 8).

“In addition, further study is necessary to confirm application of the model presented in this study for the patients with systemic diseases.”
